# Identification of exceptionally potent adenosine deaminases RNA editors from high body temperature organisms

Adi Avram-Shperling[1,2], Eli Kopel[1], Itamar Twersky[1], Orshay Gabay[1], Amit Ben-David[1], Sarit Karako-Lampert[3], Joshua J. C. Rosenthal[4], Erez Y. Levanon[1,2], Eli Eisenberg[5]*, Shay Ben-Aroya[1]*

1 The Nano Center, The Mina and Everard Goodman Faculty of Life Sciences, Bar-Ilan University, Ramat Gan, Israel, 2 The Institute of Nanotechnology and Advanced Materials, Bar-Ilan University, Ramat Gan, Israel, 3 NGS Unit Bar-Ilan University, Ramat Gan, Israel, 4 The Eugene Bell Center, The Marine Biological Laboratory, Woods Hole, Massachusetts, United States of America, 5 Raymond and Beverly Sackler School of Physics and Astronomy, Tel Aviv University, Tel Aviv, Israel

* elieis@post.tau.ac.il (EE); Shay.Ben-Aroya@biu.ac.il (SB-A)

**Data Availability Statement:** All relevant data are within the paper and its Supporting Information

## Abstract

The most abundant form of RNA editing in metazoa is the deamination of adenosines into inosines (A-to-I), catalyzed by ADAR enzymes. Inosines are read as guanosines by the translation machinery, and thus A-to-I may lead to protein recoding. The ability of ADARs to recode at the mRNA level makes them attractive therapeutic tools. Several approaches for Site-Directed RNA Editing (SDRE) are currently under development. A major challenge in this field is achieving high on-target editing efficiency, and thus it is of much interest to identify highly potent ADARs. To address this, we used the baker yeast Saccharomyces cerevisiae as an editing-naïve system. We exogenously expressed a range of heterologous ADARs and identified the hummingbird and primarily mallard-duck ADARs, which evolved at 40–42˚C, as two exceptionally potent editors. ADARs bind to double-stranded RNA structures (dsRNAs), which in turn are temperature sensitive. Our results indicate that species evolved to live with higher core body temperatures have developed ADAR enzymes that target weaker dsRNA structures and would therefore be more effective than other ADARs. Further studies may use this approach to isolate additional ADARs with an editing profile of choice to meet specific requirements, thus broadening the applicability of SDRE.

## Author summary

RNA editing alters genetic information at the RNA-level. The most common type of RNA editing is the conversion of adenosine (A) to guanosine (G), which can lead to protein diversification beyond the genomic DNA blueprint. A-to-G editing is catalyzed by members of the highly conserved ADAR protein family, which bind to specific double-stranded RNA (dsRNA) structures. The ability of ADARs to recode at the mRNA level makes them attractive therapeutic tools. The main challenge is to artificially create an editable dsRNA structure around a defined target and redirect ADAR activity to achieve high on-target

files. All RNA-seq data is provided at SRA project number: PRJNA855159.

**Funding:** This work was supported by the Israel Science Foundation (1778/20 to S.B.A, 2039/20, 231/21 to E.Y.L, 1945/18 to E.E), and the U.S-Israel Binational Science Foundation (2017/262, 2020/759 to E.E). The funders had no role in study design, data collection and analysis, decision to publish, or preparation of the manuscript.

**Competing interests:** The authors have declared that no competing interests exist.

editing efficiency. It is therefore of much interest to identify highly potent natural ADARs that could be used to induce high-level editing. Here we developed a novel approach to identify such potent enzymes. Using the baker's yeast Saccharomyces cerevisiae as a neutral testing ground, we identified two exceptionally active editors: the hummingbird ADAR2, and the mallard-duck ADAR1. We provide evidence that these birds which evolved to live with higher core body temperatures have developed ADAR enzymes that target the temperature sensitive dsRNA structures and would therefore be more potent than other ADARs. Further studies may use this approach to pinpoint additional ADAR enzymes to further broaden their applicability.

## Introduction

Over the last few years, RNA modifications have become a focus of attention in many research fields. RNA editing is unique among such modifications as it alters not only the cellular fate of RNA molecules but also the genomically-encoded sequence [1–4]. The most abundant form of RNA editing in metazoans, including the early-diverged Cnidaria (corals), is deamination of adenosines into inosines (A-to-I) [5–10]. Inosines are processed as guanosines during translation, thus making A-to-I editing a powerful endogenous means of creating inner transcriptome diversity.

A-to-I editing is mediated by members of the Adenosine Deaminase Acting on RNA (ADAR) enzyme family, which bind to specific double-stranded RNA (dsRNA) structures [11, 12]. The ADAR family consists of three enzymes: ADAR1 (also known as ADAR), ADAR2 (also known as ADARB1), and the catalytically inactive ADAR3 (also known as ADARB2). All ADARs share the catalytic deaminase domain (DD) (that is inactive in ADAR3), and one or more double-strand RNA binding domains that mediate substrate recognition. In mammals, ADAR1 is mostly responsible for editing within non-coding regions while ADAR2 is mostly responsible for the relatively few A-to-I editing that occur in protein coding regions, which may lead to non-synonymous changes in the protein products [8, 13].

This ability of ADARs to re-code makes them attractive therapeutic tools to correct genetic mutations and reprogram genetic information at the mRNA level. The prospects of utilizing CRISPR-Cas9 genome editing technology for therapeutics have attracted much attention in recent years. This approach offers a solution to chronic conditions by introducing a life-long permanent genetic modification [14–16]. However, RNA editing could be a more appropriate alternative in some cases. As opposed to DNA editing mediated by CRISPR-Cas technologies, RNA editing does not require the introduction of foreign immunogenic prokaryotic proteins. In addition, genome editing approaches raise concerns of harmful consequences of off-target events, which can permanently alter genes and disrupt cellular functions [17–21]. In contrast, RNA editing alters genetic information in a transient and reversible manner and does not affect the genomic sequence. Thus, possible adverse effects would be reversible, making this approach safer. Furthermore, the transient nature of RNA editing is more suitable for treating conditions that are not permanent, where a temporary solution is more appropriate.

Several approaches for Site-Directed RNA Editing (SDRE), utilizing ADAR's catalytic activity for RNA engineering, are currently under development. The simplest strategies use only an RNA oligo to create the dsRNA structure that would guide native ADARs to edit the target adenosine of choice [22–26]. However, other strategies use RNA oligos supplemented by engineered ADAR-like proteins [27–30]. A major challenge in this field is achieving high on-target editing efficiency [31]. It is therefore of much interest to identify highly potent natural ADARs

that could be used to induce high-level editing. This task is challenging, as it requires to discern the effect of the different ADAR enzymes and the effects of evolution of the targets themselves. Here we developed a novel approach to screen and identify such potent ADAR enzymes, using as a neutral testing ground the yeast *Saccharomyces cerevisiae*, an organism whose origin precedes the emergence ADARs and therefore does not possess an endogenous capacity for A-to-I RNA editing. We exogenously expressed different heterologous ADARs to reveal their distinct editing potential and identified two exceptionally active A-to-I editors: the hummingbird ADAR2, and primarily the mallard-duck ADAR1, whose core body temperature is 40–42˚C. We provide evidence that these birds which evolved to live with higher core body temperatures have developed ADAR enzymes that target weaker dsRNA structures and would therefore be more potent than other ADARs (compared at equal temperatures).

## Results

### Exogenous expression of heterologous ADARs in yeast differently affects their growth rate

Yeast cells do not encode the ADAR proteins, and thus are not capable of A-to-I RNA editing and are not adapted to it. Accordingly, imposing editing on these cells by introduction of exogenous ADAR enzymes results in impairment of their growth rate [32]. We thus proposed to screen different ADAR enzymes using growth impairment as a probe for editing activity.

ADARs contain dsRNA Binding Domains (RBDs) that recognize RNA secondary structures surrounding the target adenosine. The stability of these structures is temperature-dependent [33, 34], and weaker (less paired) structures are generally expected at higher temperatures. We have therefore speculated that species evolved to live with higher core body temperatures may have developed ADAR enzymes that target weaker dsRNA structures.

Endothermic vertebrates maintain a fixed core body temperature. This body temperature is between 35 and 42˚C, which is 10–15˚C higher than the typical body temperature of ectothermic vertebrates [35]. We have therefore looked at ADAR1 and ADAR2 orthologs from five species–two mammals and two birds, and one ectotherm invertebrate, that inhabit different environmental niches: (1) Human (*Homo sapiens*), whose ADARs are commonly used in base editing research [32, 36, 37]; (2) Squid (*Loligo opalescens*), an ectotherm invertebrate living in diverse temperatures, which was shown to have extraordinarily high levels of A-to-I editing in coding sequences [38–40]; (3) The marine mammal, Orca whale (*Orcinus orca*), with core body temperature of ~38˚C [41]; (4, 5) Hummingbird (*Calypte anna*) and Mallard duck (*Anas platyrhynchos*), two birds with relatively high core body temperatures of ~40˚C and ~42˚C respectively, the warm end of the endothermic spectrum [42, 43].

We cloned each of these ten heterologous enzymes into a *URA3* marked plasmid that enables their expression under the galactose inducible promoter (*GAL1*p) (Fig 1A and 1B). Next, we immunoblotted protein lysates samples from yeast cells carrying these plasmids and were able to validate the expression of most enzymes, except hbADAR1, owADAR1 and sqADAR1, which were most probably eliminated by the proteasome ubiquitin system or autophagy (Fig 1C). These strains were then serially diluted and spotted on galactose- and glucose-containing media (ADAR expression is on and off, respectively) and incubated at 30˚C. As shown in Fig 1D, the strongest growth impairment effect was seen upon inducible expression of Orca whale ADAR2 (owADAR2), Hummingbird ADAR2 (hbADAR2), and Mallard duck ADAR1(mdADAR1).

To further confirm and quantitate the growth defect of these strains, we measured the optical density of liquid cultures over 22 hours (in 30-minutes intervals) for these three affected strains, as well as two reference strains carrying the squid ADAR2 (sqADAR2) and Human

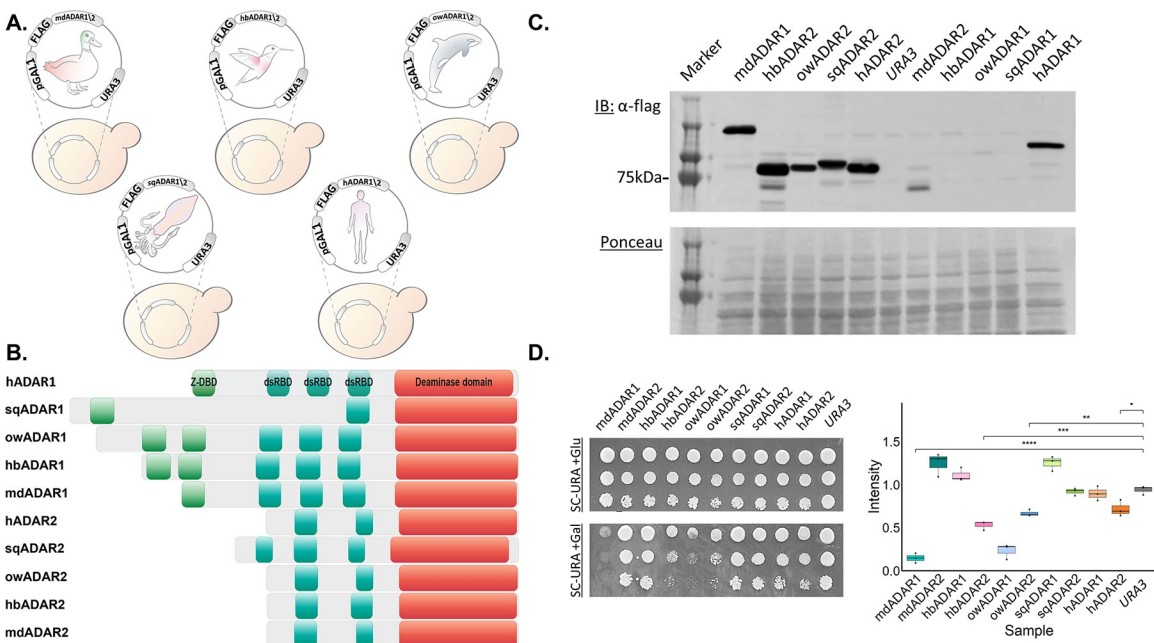

**Fig 1. Exogenous expression of heterologous ADARs in yeast differently affects their growth rate (A) Schematic representation of the yeast-based expression system.** The coding sequence of heterologous ADAR enzymes N-terminally fused to a FLAG-tag were cloned into a *URA3* marked plasmid under a galactose inducible promoter (*GAL1*p). ADARs were originated from Mallard duck (*Anas platyrhynchos*; mdADAR1/2); Hummingbird (*Calypte anna*; hbADAR1/2); Orca whale (*Orcinus orca*; owADAR1/2); Squid (*Loligo opalescens*; sqADAR1/2); and Human (*Homo sapiens*; hADAR1/2). **(B) Multiple alignment of the heterologous ADARs used in this study.** The ADAR enzymes contain a catalytic deaminase domain (DD, red), two or three double stranded RNA binding domains (dsRBD, blue), and, in ADAR1 only, one or two Z-DNA binding domains (Z-DBD, green). Domains were identified using PROSITE database. All schemes are to scale. **(C) Western blot analysis confirms the expression of the ADAR proteins in yeast.** Logarithmically growing cells harboring the plasmids described in A were grown in minimal media (SC-URA) supplemented with raffinose (Raf) (SC-URA+Raf) (ADAR expression is off). 2% Galactose (Gal) was then added for 6 hours, to induce the ADAR genes expression. Cell lysates from the SC-URA+Gal media were separated by SDS-PAGE and immunoblotted with anti-FLAG (α-FLAG) antibody. Cells harboring an empty *URA3* marked plasmid were used as a negative control (*URA3*). Ponceau staining was used as a loading control. **(D) The inducible expression of specific heterologous ADAR enzymes impairs cells growth.** *Left*: 10-fold serial dilutions of the indicated strains were spotted on Glucose (top, Glu, ADAR expression is off) and Galactose containing media (bottom, Gal, ADAR expression is on). Plates were incubated at 30°C for 40h. Cells carrying an empty *URA3* plasmid were used as a control. *Right*: Quantitation of growth impairment. Growth was assessed using a digital image of a drop (second dilution) to generate an estimate of the growth on SD-Gal relative to SD-Glu based on pixel density. Whiskers represent the minimum and maximum values measured from three independent experiments. Stars above the boxplots indicate a statistically significant difference between the means of two samples (ns: $p > 0.05$; *: $p <= 0.05$; **: $p <= 0.01$; ***: $p <= 0.001$; ****: $p <= 0.0001$).

ADAR2 (hADAR2) which have shown a weaker effect (Fig 2A). The growth curves confirmed that the inducible expression of all five ADAR enzymes resulted in differential growth impairment, with a stronger effect for owADAR2, hbADAR2, and the most discernible effect in cells overexpressing mdADAR1. No effect was detected when the same strains were grown in a glucose-containing media, where no ADAR protein expression is detected (Fig 2B).

## The impaired growth rate of yeast cells expressing heterologous ADARs is the result of editing within yeast transcripts

To verify that the exogenously induced ADAR enzymes are catalytically active in the yeast cell, we sequenced RNA samples from the strains described in Fig 2 and looked for evidence of A-to-I editing. In addition, we looked at three negative controls, cells harboring an empty *URA3*-marked plasmid, and cells expressing catalytically inactive mdADAR1 and hADAR2,

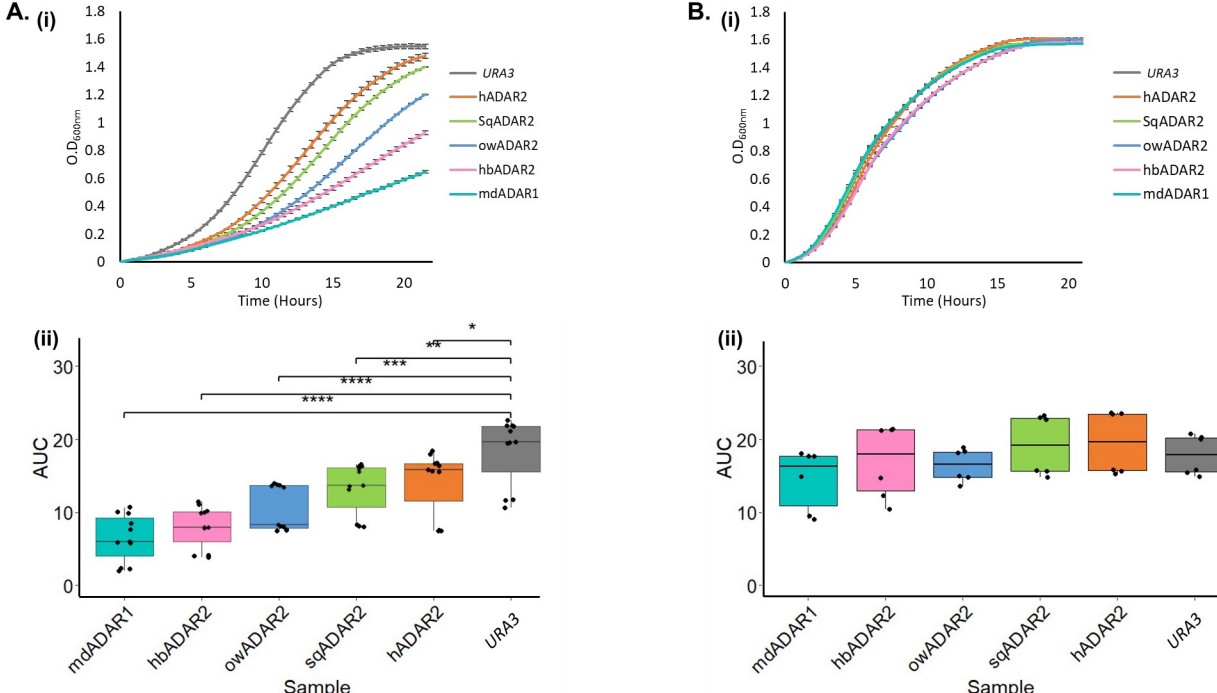

**Fig 2. The expression of heterologous ADARs in yeasts impairs growth differently. (A, B) (i)** Logarithmic cells carrying hADAR2, sqADAR2, owADAR2, hbADAR2, and mdADAR1, were grown in SC-URA+Raf. Cells were then normalized to an optical density at a wavelength of 600nm (O.D$_{600nm}$) of 0.1 and transferred to SC-URA+Gal (A) and SC-URA+Glu (B) (ADAR expression on and off respectively). Growth was assessed in 96 wells plate using a TECAN microplate reader, by measuring the O.D$_{600nm}$ in 30 minutes intervals for 22 hours. Cells harboring an empty *URA3* marked plasmid were used as a control. Error bars show the standard deviation between three independent experiments. **(ii)** The growth in A and B was quantitated by calculating the area under the curve (AUC). Whiskers represent the minimum and maximum values measured from three independent experiments. Stars above the boxplots indicate a statistically significant difference between the means of two samples (ns: $p > 0.05$; *: $p < = 0.05$; **: $p < = 0.01$; ***: $p < = 0.001$; ****: $p < = 0.0001$).

(mdADAR1-E619A and hADAR2- E396A) in which a critical glutamic acid was converted to alanine. A similar mutation was shown to inhibit catalytic activity in hADAR1 [44].

Inosines are read by the reverse-transcriptase and the following sequencing protocol as guanosines. Thus, editing detection tools look for A-to-G mismatches between the reference genome and the RNA-seq reads. As a control, we looked at the number of mismatches of different types. Since our RNA-sequencing libraries were not strand-specific, there are only six possible types of mismatches (e.g., T-to-C mismatches are indistinguishable from A-to-G ones). First, we used the RES-scanner pipeline [45] followed by multiple testing correction to identify mismatch sites de-novo (see Methods). We found that the number of non-A-to-G substitutions were similar for all samples, while the number of A-to-G mismatches (presumably induced by A-to-I editing) increased substantially upon ADAR induction. For samples containing active ADARs 74–99% of the identified sites were A-to-G mismatches, and the number of these mismatches was at least 25-fold higher than that observed in control samples (Fig 3A). The number of editing sites per sample varied from 597 for hADAR2 (~25-fold increase from the baseline A-to-G mismatch level shown in the control samples) to 113,672 for mdADAR1 (~5000-fold increase from the baseline), which has the highest body temperature of the species screened. The detected A-to-G sites exhibit the familiar ADAR motif [46, 47] (Fig 3B). We thus attributed these mismatches to A-to-I RNA-editing events. The majority of these editing sites resides within protein coding sequences, and the distribution of editing levels is skewed towards lower values, as expected (see S1 Fig).

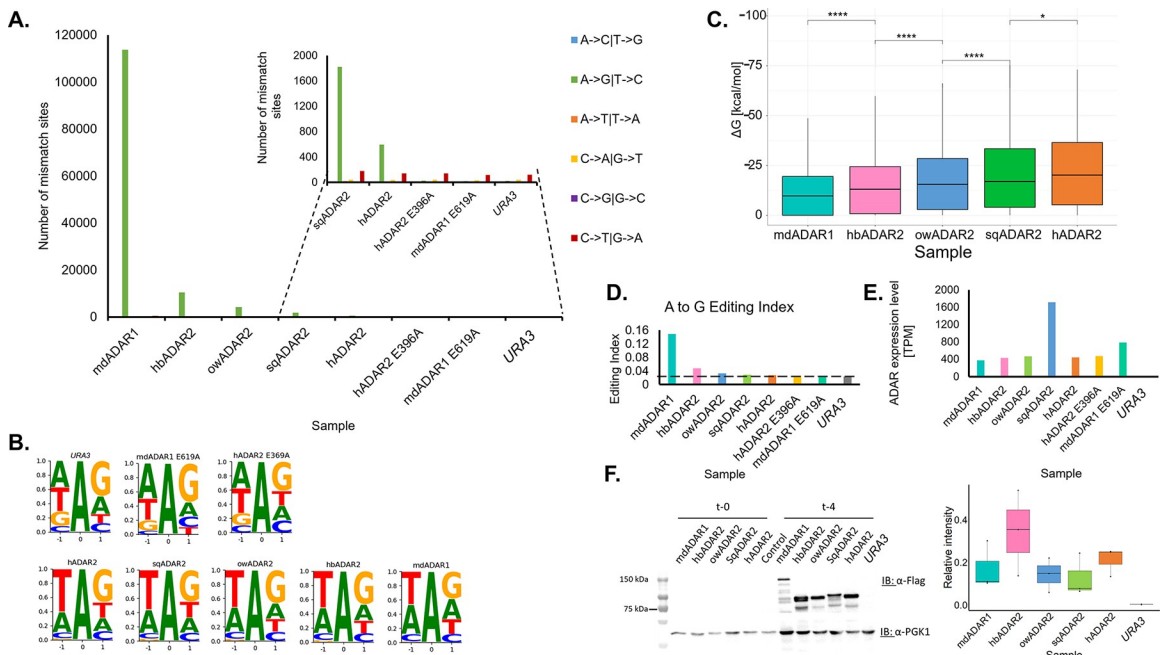

**Fig 3. Editing activity in ADAR-expressing yeast strains is strongest for mallard duck ADAR1. (A)** Res-scanner detection of editing sites (Methods) for five yeast strains expressing different active ADAR enzymes results in varying amounts of sites, spanning two to three orders of magnitude. Note that due to the large numbers for mdADAR1 (113,672 sites, in a 12Mbp genome) and hbADAR2 (10,443), the results for the other strains are invisible in the main graph, and thus these results are plotted again in the inset. The (three rightmost) control strains show no excess of A-to-G mismatches, attesting for no editing activity as expected. **(B)** Distribution of the neighboring nucleotides to the predicted editing sites reveals the familiar editing motif (mostly a depletion of G upstream) for all strains expressing active ADARs, but not for the control ones. **(C)** Thermodynamic stability of the predicted secondary structures surrounding the detected sites (Methods). Lower (more negative) $\Delta G$ indicates a more stable structure. Note that no structure was found for 113, 379, 939, 2526 and 32197 sites, for the five organisms, respectively. Their $\Delta G$ was set to zero. **(D)** Genome-wide editing index (Methods) for the five active ADAR strains and three control ones. The dashed line represents the base-line noise level. **(E-F)** ADAR expression levels (TPM) (E) and protein levels (F) for the different strains do not account for the wide differences in the number of sites detected. Logarithmically growing cells carrying the indicated plasmids were grown in SC-URA+Raf (ADAR expression is off). 2% Gal was then added for 4 hours, to induce the ADAR genes expression. Cell lysates from the SC-URA+Raf and SC-URA+Gal (t-0 and t-4 respectively) were separated by SDS-PAGE and immunoblotted with anti-FLAG (α-FLAG. Anti-3-phosphoglycerate kinase 1 (α-PGK1) was used as a loading control. Relative intensity was measured as pixel density for each sample with consideration of the loading control. Protein levels were quantitated relative to the loading control in three independent experiments. Whiskers represent the minimum and maximum values measured from three independent experiments. Stars above the boxplots indicate a statistically significant difference between the means of two samples (ns: p > 0.05; *: p < = 0.05; **: p < = 0.01; ***: p < = 0.001; ****: p < = 0.0001).

We wanted to further test our hypothesis that the increased number of editing sites using ADARs originating from species with higher core body temperatures is due to targeting of weaker dsRNA structures by these enzymes. We thus looked at the thermodynamic stability of the secondary structures surrounding the editing sites detected per strain (Methods). As expected, enzymes editing more sites have also targeted sites with a lower stability (Fig 3C). In particular, the structures edited by mdADAR1 are significantly less stable that those edited by hADAR2, in agreement with our proposed model.

In addition, in order to obtain a quantitative comparison between the different enzymes we used the editing index approach (see Methods), which is less sensitive but more robust [48]. Briefly, the editing index represents the fraction of nucleotides expressed from genomically-encoded adenosines that has been edited. Here, we have calculated the index across the full yeast genome. In the *URA3* empty plasmid sample and the two inactive ADAR samples, the measured editing index was 0.023–0.025%, representing the baseline noise level (mainly due to

genomic polymorphism sites). The samples expressing active ADARs exhibited elevated index values, as expected, and the increase correlated to the number of de-novo sites reported above (Fig 3D). We also quantified the RNA and protein expression levels of the ADAR enzyme in each sample (Fig 3E and 3F). While there are variations between the samples, the high editing seen for mdADAR1 is not explained by higher RNA or protein levels, and likely represents the intrinsic potency of this enzyme. Notably, for all ADAR enzymes tested, the majority of detected sites are also detected with mdADAR1 (S2 Fig).

To further validate these results, we sequenced again strains including empty vectors, inactive mdADAR1-E619A and active mdADAR1, five replicates each. The number of A-G/ T-C mismatch sites detected by RES scanner was 6±2 and 18±7 for the empty plasmid control strain and the mutated mdADAR1, respectively, and increased to 240,600±44,700 upon induction of the active mallard duck enzyme. Similarly, editing index was similar for the empty plasmid control strain and the mutated mdADAR1 (0.0154±0.0006% and 0.0161 ±0.0008%, respectively), but increased substantially for the active enzyme– 0.313±0.050% (S3 Fig).

Altogether, we found 471,784 distinct adenosine positions that were identified as edited in at least one (out of five) mdADAR1 samples (S3 Table - https://docs.google.com/ spreadsheets/d/1wXGKUPB39t4P-4AiXq9ZGwfxsR_NrHcw/edit?usp=share_link&ouid= 102710005033051837999&rtpof=true&sd=true). That is, we observe editing in about 1 in 16 A:T sites in the whole yeast genome (regardless of the expression level of the locus and strand in which the adenosine resides). Of these, 85,726 sites (~18%) reoccur in all five samples (S4 Fig). Most of the sites were weakly edited, but 79,687 sites (~17%) have shown editing levels of 10% or more in at least one of the samples. Overall, the index value of ~0.3% means that ~1/300 of the adenosines expressed in the transcriptome is converted to inosine. These findings imply that in the absence of long-time adaptation in the ADAR-naïve yeast genome, a sizable amount of adenosine sites are located within preferable substrates for mdADAR1 activity. These results, in agreement with the observations from cell growth assay, demonstrate the potency of mdADAR1.

mdADAR1 and hbADAR2 carry a normal number of dsRBDs (Fig 1B). Their sequences exhibit multiple differences with respect to ADAR enzymes from other taxa, and it is not easy to pinpoint the source for their exceptional editing potency. To identify the domains that contribute to the unique editing capability of mdADAR1 we carried out domain deletion experiments, expressing under GAL1p constructs that include mdADAR1 with its (a) first, (b) first and second, or (c) all three dsRBDs deleted (Fig 4A). In addition, we carried out domain swapping experiments, testing the growth of strains carrying the following hybrids: (i) mdADAR1 combined with the human-dsRBDs1,2,3 (mdADAR-DD-hRBD1-3), and (ii) human ADAR1-DD combined with the md-dsRBDs1,2,3 (hADAR1-DD-mdRBDs1-3) (Fig 4B). The results show that the growth rate of strains where some or all of the dsRBMs were deleted was similar to the wt control, implying that the combination of the DD and all three dsRBDs is important for mdADAR1 editing. Furthermore, the growth of the hADAR1-DD-mdRBDs1-3 strain was also comparable to the wt. In contrast, growth was partially impaired in the mdA-DAR-DD-hRBD1-3 strain, even though not as much as for the full mdADAR1 strain (Fig 4C). These results imply that the mdRBDs are partially replicable by the hRBD2-3, and that the mdADAR1 catalytic domain has a larger contribution to its potency in yeast.

To conclude, introduction of active ADAR proteins into the ADAR-naïve yeast cells resulted in editing of thousands of sites, which can be readily probed by growth impairment. Of the enzymes tested, mdADAR1 is by far the most potent one, resulting in hundreds of thousands of editing sites and a severe growth impairment.

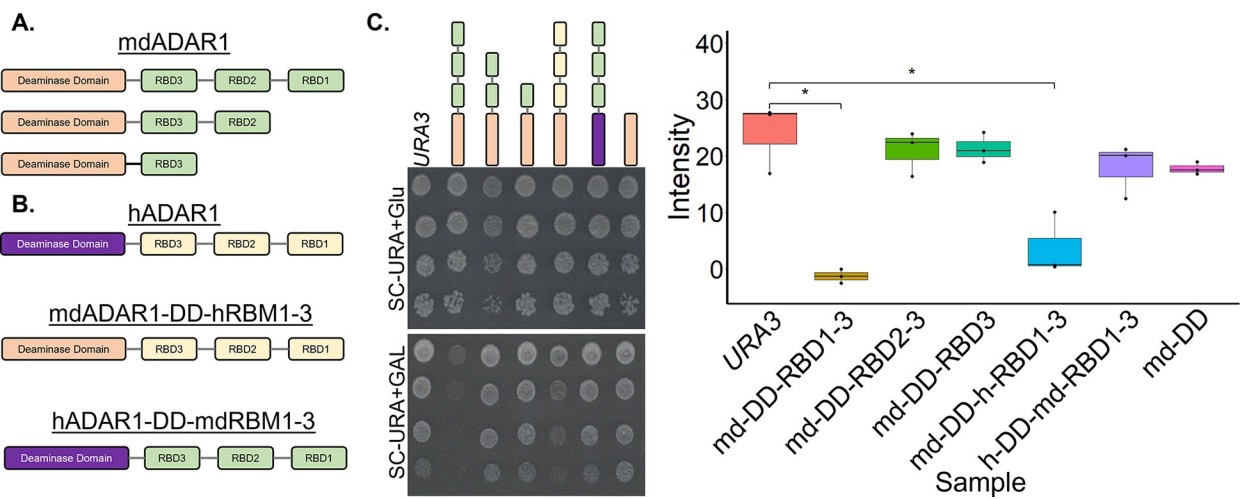

**Fig 4. mdADAR1 potency is mainly due to its catalytic domain. (A, B)** Schematic representation of the full length mdADAR1, the shorter deletion (A), and the domain swapping constructs tested (B). C. 10-fold serial dilutions of the indicated strains were spotted, and growth was quantitated as described in 1D. Plates were incubated at 30˚C for 40h. Cells carrying an empty *URA3* plasmid were used as a control.

## dsRNA substrate recognition by heterologous ADARs in yeast are differentially disturbed by temperature change

As temperature decreases, shorter and imperfect dsRNA helices are stabilized, resulting in an increased number of potential ADAR substrates. All the experiments above were performed at 30˚C. We therefore wanted to examine the relative potency of the enzymes and the temperature-dependence for each one of them over an extended temperature regime. We focused on the yeast strain expressing: (1) mdADAR1, and (2) hbADAR2, the first and second most potent ADARs, with the greatest effect on the yeast transcriptome and growth rate, and (3) hADAR2 as a reference strain with the least effect. These strains were serially diluted and spotted on glucose- or galactose-containing media and incubated at 25˚C, 30˚C and 34˚C. mdADAR1 growth was severely impaired at all three temperatures, but in the case of hbADAR2 the strong growth impairment at 25˚C was partially and gradually restored when ramping up the temperature to 30˚C and 34˚C (Fig 5A). On glucose, when ADAR expression is off, temperature shift had minimal effect and growth was comparable to the control strain at all temperatures. Immunoblotting of protein samples originated from these cells shows that ADARs' protein levels was not affected by the temperature shift, ruling out the possibility that the effects are the results variation in protein levels (Fig 5B).

To complement these data, we sequenced the RNA and quantified A-to-I editing in these samples as described above, looking at three replicates each time. Using RES-scanner to detect mismatches de-novo, we found that for hADAR2 and hbADAR2 the editing signal increases substantially with decreasing temperature (Fig 6A). In particular, for hADAR2, where the effect of editing on viability is the smallest of all three enzymes, the number of detected sites increases by more than an order of magnitude upon decreasing the temperature from 34˚C to 25˚C. In mdADAR1 the behaviour is more complex, presumably due to the dramatic effect of high-level editing on viability, even at the highest temperature. The editing index shows a similar pattern (Fig 6B). Here too, we verified that the increase in the number of sites edited by hADAR2 with lowered temperature is due to the stability of the secondary structures surrounding the editing sites. For each of the sites detected, we analysed the surrounding structures predicted for 30˚C (Methods) and found that lowering the temperature allows for editing of less stable structures, as expected (Fig 6C). In summary, the ambient temperature has a

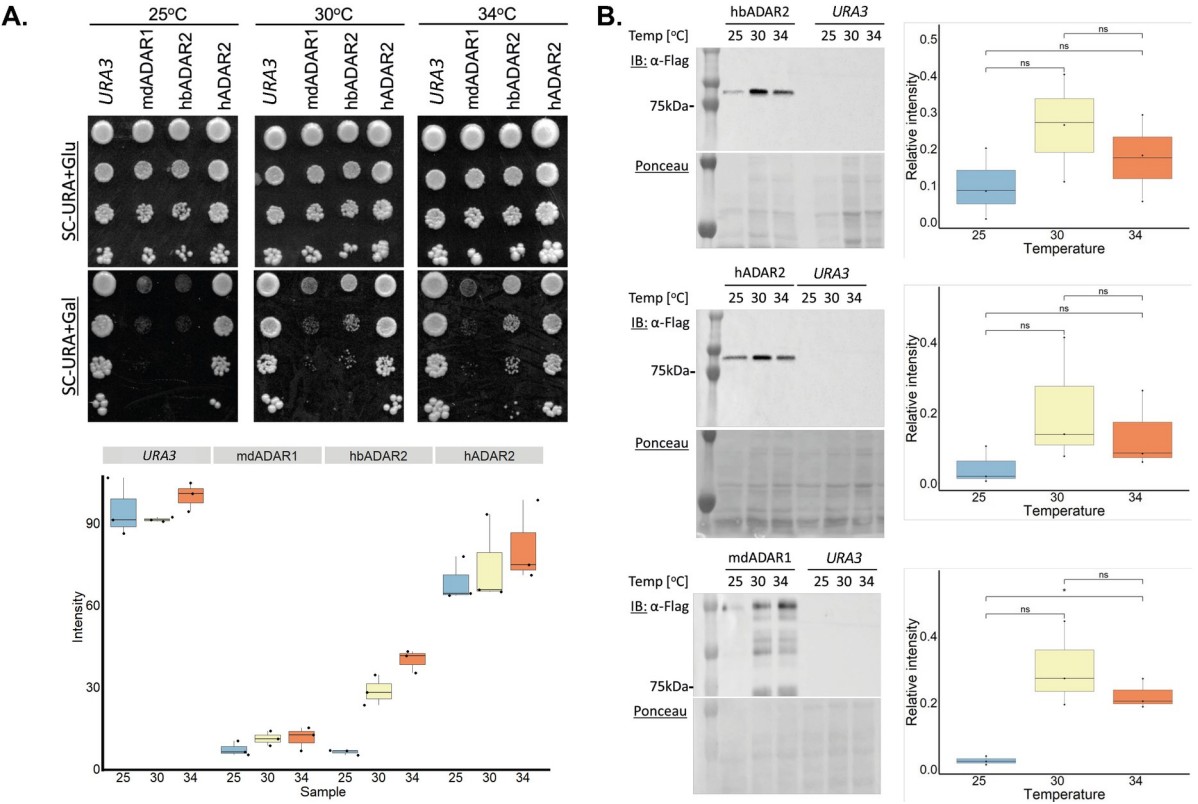

**Fig 5. The growth of yeast cells carrying heterologous ADARs is differentially affected by temperature change. (A)** Similar to 1D, this time plates were incubated at 25°C, 30°C or 34°C for 40h. **(B)** Protein levels of the ADAR proteins in A are not significantly affected by the temperature shift. Similar to 1C, this time samples from the SC-URA+Gal media were incubated at 25°C, 30°C or 34°C for 6 hours. Cells harboring an empty *URA3* marked plasmid were used as a negative control (*URA3*). Ponceau staining was used a loading control. Protein levels were quantitated relative to the loading control in three independent experiments. Stars above the boxplots indicate a statistically significant difference between the means of two samples (ns: p > 0.05; *: p < = 0.05).

strong effect on editing activity, but the mdADAR1 remains the most potent enzyme even at temperatures that are far remote from its native environment.

Finally, we looked at editing by mdADAR1 in mammalian systems. We have expressed hADAR2, hbADAR2 and mdADAR1 in human HeLa cells, and compared the global editing using the Alu editing index (Methods) [48]. Here too, hbADAR2 and mdADAR1 have shown an increased level of editing, much larger than the one seen for over-expression of hADAR2 (S5 Fig).

## Discussion

Yeast cells have been previously used as a model system to study ADAR functionality and preferences. These studies have focused on human ADARs and identified a novel set of editing substrates [32]. Furthermore, co-expression of human ADARs with selected yeast substrates have allowed for monitoring the effects of either mutations within ADAR domains or the target dsRNA structure, on the editing levels [37, 49, 50]. Here we use the yeast model system to compare a range of ADAR proteins. We detected extensive transcriptome-wide A-to-I changes, accompanied by noticeable effects on cell growth which imply a biological impact for the editing events. We thus conclude that yeast cells can be used as a screening platform to reveal the inherent potential of selected ADARs. Using this approach, we singled out the

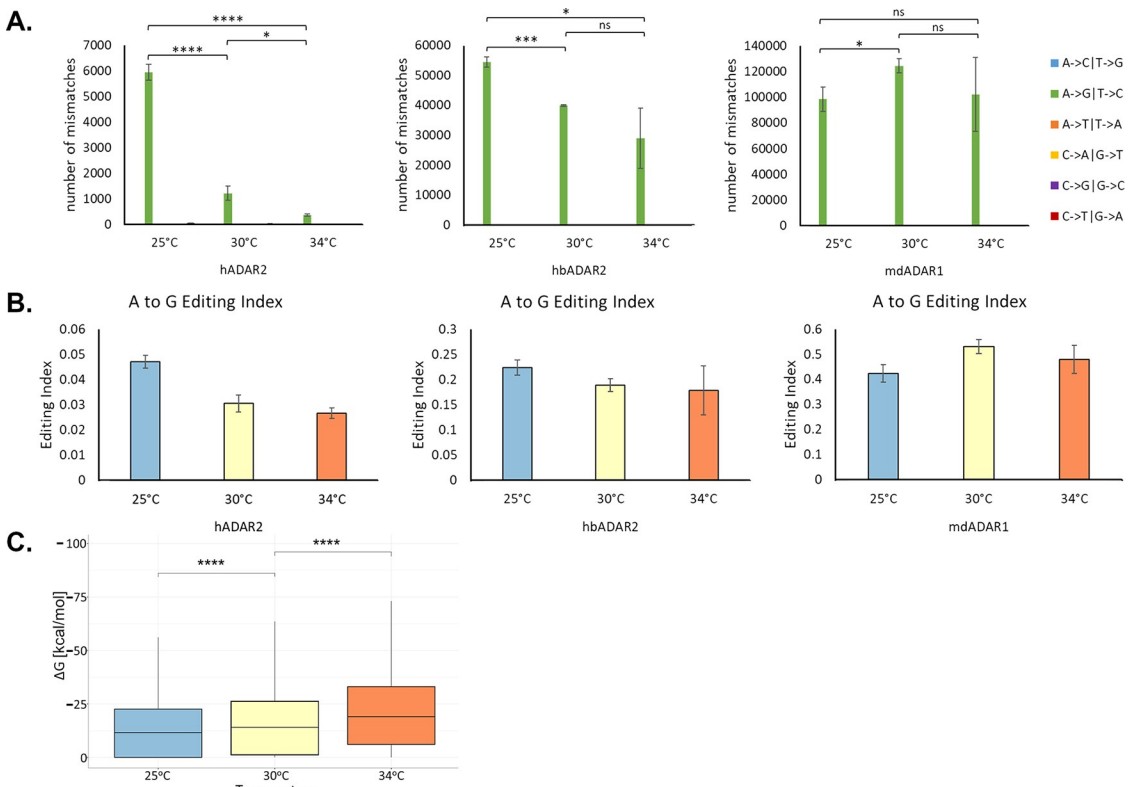

**Fig 6. Temperature dependence of editing in ADAR-expressing yeast strains. (A)** Res-scanner detection of editing sites (Methods) demonstrates a clear negative correlation between the growth temperature and editing activity for hADAR2 and hbADAR2. The strain expressing mdADAR1, where the impact of editing is much more extensive and severe, exhibits a more complex behavior. **(B)** The same picture emerges looking at the genome-wide editing index. **(C)** Thermodynamic stability of the predicted secondary structures surrounding the detected sites edited by hADAR2 (Methods). For each temperature, sites detected in at least once replicate were included. For the sake of comparison, all structures were calculated for 30°C. Lower (more negative) ΔG indicates a more stable structure. Note that no structure was found for 2727, 537 and 111 sites, for T = 25°C, 30°C and 34°C, respectively. Their ΔG was set to zero. (ns: $p > 0.05$; *: $p < = 0.05$; **: $p < = 0.01$; ***: $p < = 0.001$; ****: $p < = 0.0001$).

mallard duck ADAR1 as an exceptionally potent A-to-I editor, capable of editing up to hundreds of thousands of sites in the yeast transcriptome.

ADARs substrate recognition mainly depends on a duplex RNA secondary structure surrounding the target adenosine, which is recognized by ADARs' dsRNA Binding Domains (RBDs). dsRNA secondary structure is affected by the ambient temperature [33, 34], as one generally expects tighter dsRNA structures at lower temperatures. Thus, editing is expected to depend on temperature. Temperature-dependence of editing levels has been demonstrated in Drosophila [34, 51, 52]. Furthermore, several studies have established that the response of RNA editing to temperature changes may act as a means for adaptation and acclimation [39, 53]. An intriguing example is presented by the ectothermic cephalopods which show extensive RNA editing [38, 40, 54]. Nonsynonymous editing site in the potassium voltage-gated channel subfamily A member 1 (Kv1.1) was shown to depend on the ambient temperature at which the octopus species lives, suggesting RNA editing contribution to adaptation [39]. Accordingly, maintaining editing at the desired level may require adaptation of the editing substrate to the ambient temperature through mutations that modify the RNA structure stability [55].

In parallel, homoeothermic species could also adapt the editing enzymes to their body temperature, changing their dsRNA binding and the catalytic activity to compensate for inter-species

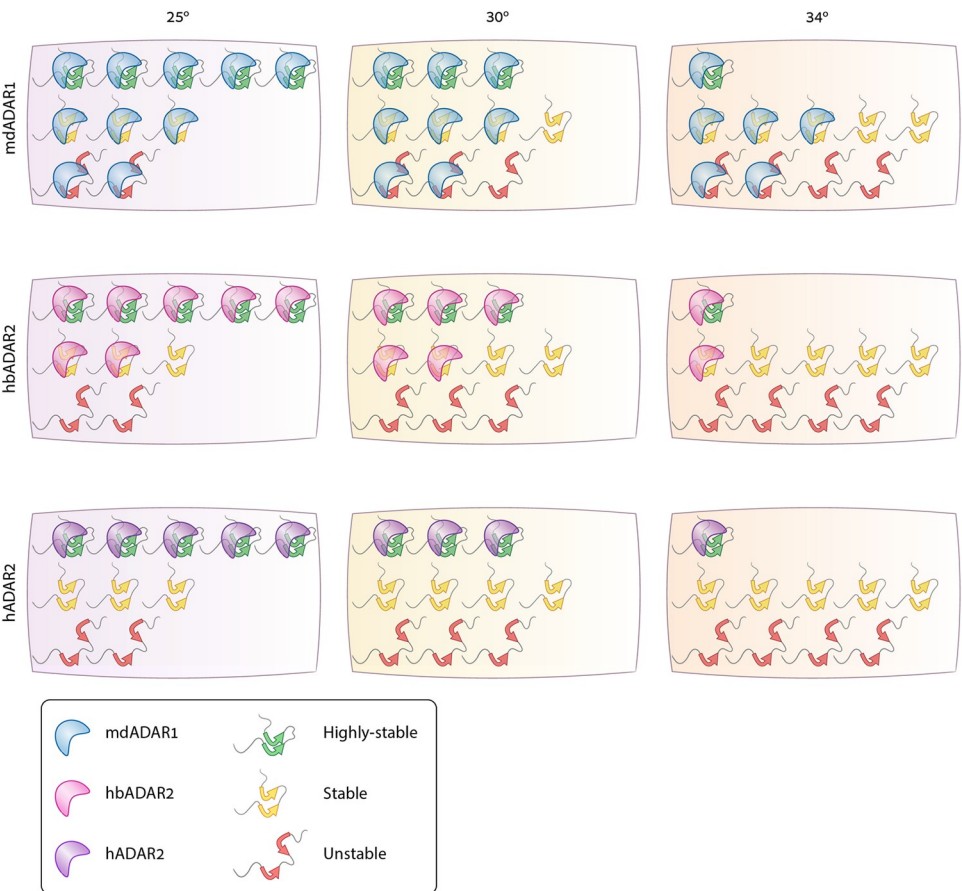

**Fig 7. Possible model for the hyper A-to-I editing resulting from the exogenous expression of mdADAR1 in yeast.**
mdADAR1 and hbADAR2 originated from birds with core temperature of 42°C and 40°C, the warm end of the endothermic spectrum, have probably evolved to better recognize partially destabilized dsRNA structures. Their exogenous expression in yeast stimulated the highest association with dsRNA structures and editing activity in all temperatures when compared to hADAR2 evolved at lower body temperature. Temperature increase leads to a shift in the distribution of structures, with less tight ones (green) and more partially destabilized ones (red), which is reflected in the total number of edited sites and a partial growth restoration of growth and viability in cells carrying the hbADAR2, to some extent, and mostly hADAR2.

differences in body temperatures. We therefore suggest that species evolved to live with higher core body temperatures have developed ADAR enzymes that target weaker dsRNA structures and would therefore be more potent than other ADARs (compared at equal temperatures).

The model we have in mind is as follows: Temperature affects the stability of the dsRNA structures themselves but also the binding of the enzyme to a given structure. ADARs must edit tight dsRNAs to avoid false alarming the innate immune system. Enzymes evolved at high body temperature must be active against the dsRNA structures that sufficiently tight, despite the lower enzyme-substrate affinity dictated by the high temperature. Accordingly, they are expected to be more sensitive than enzymes evolved at lower body temperatures, when compared at the same ambient temperature. In parallel, changing the temperature also increases the repertoire of tight dsRNAs, leading to an increase in the number of edits for a given enzyme (Fig 7).

We currently don't know what mechanism might underlie the elevated editing by the identified ADARs. Their RNA expression level and western blot analysis rules out the most obvious

explanation, that they are highly abundant. This implies that their high editing activity is likely due to the intrinsic potency of the enzymes. They carry a normal number of RBDs, but their sequence shows multiple differences when aligned with ADARs from other taxa. Based on deletion and domain swap experiments in mdADAR1, we conclude that the existence of all its three dsRBDs is necessary, but they are not essentially different from hADAR1 dsRBDs, and the major contributor to the editing activity of mdADAR1 is the DD. Future mutagenesis experiments combined with functional and validation studies will help shed light on this question.

The severe growth impairment observed in mdADAR1 yeast strains highlights the potential grave impact of over-editing. This raises the question of how does the mallard itself tolerate its potent native ADAR enzyme. Possibly, this enzyme is potent only when expressed at cold temperatures, but does not edit extensively at 42˚C. Alternatively, the mallard duck genome may have evolved to avoid dsRNA structures that could be targeted by ADAR.

For hADAR, we observed a clear increase of editing activity with lowering the temperature. Extrapolating to the native temperature of this enzyme, 37˚C, it seems that there would be very little editing activity in the yeast transcriptome by hADAR2. This suggests that the existence of hADAR2 editing sites have required special adaptation of the surrounding sequence to create sufficiently tight dsRNA structures that are not present, or at least are rare, in the yeast transcriptome.

ADAR-based base-editors may prove to be an attractive RNA engineering and therapeutic tool. Using the editing-naïve system we have demonstrated the variability in activity across ADAR enzymes from different species, and identified the mallard duck ADAR as an exceptionally potent editing enzyme. Further studies may use this approach to pinpoint additional ADAR enzymes of various desired editing profile, to further broaden the applicability of ADAR-based systems.

## Material and methods

### Yeast strains

Unless otherwise stated, all the strains used in this study are isogenic to BY4742 [56]. The relevant genotypes are presented in S1 Table.

### Growth conditions

Yeast cells were grown in synthetic complete medium (SC; 0.17% yeast nitrogen base, 0.5% $(NH_4)_2SO_4$, and amino acids), supplemented with either 2% glucose (SC+Glu), raffinose (SC +Raf), or galactose (SC+Gal). Unless otherwise stated, cells were grown at 30˚C. For logarithmic culture, cells were grown for 16–18 hours and then back diluted 10x with fresh media and allowed to grow for the indicated time. Standard YEP medium (1% yeast extract, 2% Bacto Peptone) supplemented with 2% dextrose (YPD) was used for nonselective growth. 2% Bacto Agar was added for solid media.

### HeLa cells

Cells were cultured in Dulbecco's Modified Eagle Medium (DMEM) (Biological Industries, 01-055-1A) supplemented with L-Alanyl-L-Glutamine (Biological Industries, 03-020-1B), Foetal Bovine Serum (FBS) European Grade Heat Inactivated (Biological Industries, 04-127-1A) and Penicillin-Streptomycin-Nystatin Solution (Biological Industries, 03-032-1B). mdADAR1, hbADAR2 and hADAR2 were synthesized by Twist Bioscience with an N' Flag (DYKDDDDK) and cloned into pTwist-CMV promoter-Puromycin selection plasmids. Codons were optimized using Twist codon optimization tool. Transfection was carried out

using jetPRIME reagent (Polyplus Transfection). Puromycin (InvivoGen ant-pr-1) was used for selection.

## Plasmids

Heterologous ADAR genes (mdADAR1, mdADAR2, hbADAR1, hbADAR2, owADAR1, owADAR2, sqADAR1, sqADAR2, hADAR1, hADAR2) were synthesized by Integrated DNA technologies (IDT) with an N' Flag (DYKDDDDK) and a C' x6HIS tags (S2 Table). Codon optimization was preformed using IDT codon optimization tool. hADAR2 was cloned into pYES-DEST52 (Invitrogen #12286019) using the gateway system. All other ADARs were PCR amplified with flanking homology to the pYES-DEST52 vector and cloned by in-vivo homologous recombination in yeast. pYES-DEST52-hADAR2 was digested using BamHI-HF (New England Biolabs, NEB-R3136S) and co-transformed with PCR fragments into BY4742.

Plasmids were purified from yeast using Zymoprep Yeast Plasmid Miniprep I (Zymo Research, ZR-D2001) and transformed into DH5α. Plasmids were purified from DH5α using Minipreps DNA Purification (Promega A1460). Cloning was verified by sequencing of 3' and 5' integration sites.

Chimeric ADARs were generated by PCR amplification of the selected domains which were inserted into the pYES-DEST52-mdADAR1 digested with BglII (New England Biolabs, NEB- R0144S) by in-vivo homologous recombination in yeast.

## Protein extraction and immunoblotting analysis

Logarithmically growing cells were washed twice with water and resuspended in 300μl ice-chilled buffer B60 (50ml HEPES pH7.3 1M, 1ml Triton, 4.32g β-glycerophosphate, 5.92g potassium acetate, and ~600ml DDW) containing 1:200 protease inhibitor cocktail (Merck MBS5391341ML), and 1.5g of glass beads was added. The tubes were vortexed six times for 2min with 2min intervals on ice. After 2 min on ice, the lysate was centrifuged for 20min at 18,000×g at 4°C.

Protein samples were normalized using 1:5 Bradford solution to a total volume of 25μl (30μg protein, 5μl leamlli (4ml 1.5M Tris-Cl pH6.8, 10ml glycerol, 5ml β-mercaptoethanol, 2g SDS (sodium dodecyl sulfate/ sodium lauryl sulfate), 1ml 1% bromophenol blue) and DDW). Proteins were separated in 10% SDS-PAGE, transferred to nitrocellulose membrane (Bio-Rad #1704271), and then incubated with α-Flag-HRP (1:3000, Abcam, AB-ab1238). ImageJ software (National Institutes of Health, Bethesda, MA, USA) was used to quantitate the Western blots data.

## Cell growth assays

To measure cells viability, exponentially growing cells were normalized to a density of $1x10^6$ cells/ml into a 96-well plate containing selective medium supplemented with galactose to induce ADAR genes expression and incubated for 24h at 30°C while shaking. Cell growth was then determined by measuring the absorbance at 600nm in 30min intervals using Tecan Spark 10M multimode microplate reader. Area under the curve (AUC) was calculated using "Growthcurver" package in R.

Cell growth following viability drop tests were quantitated by scanning the plates (Epson perfection V370). ImageJ software (National Institutes of Health, Bethesda, MA, USA) was used to quantitate pixel density.

## RNA purification and RNAseq

Logarithmic cells grown in minimal medium supplemented with galactose to induce heterologous ADARs expression were incubated at indicated temperatures for 6h. RNAs were purified

using Yeast RNA Purification Kit (epicentre MPY03100). PerfeCTa DNAse I (Quanta, 95150-01K) was used to remove DNA contamination. RNA was stored in RNAse-free water at -80˚. RNA integrity was assessed by Tapestation (Agilent Technologies, Inc).

### Library preparations

For library preparation (Cohort1) NEBNext RNA ultra II RNA library preparation kit (NEB, Ipswich, MA) was used from 1 ug of starting material. All RNA samples underwent PolyA selection following the manufacturers' protocols, with a 7 PCR cycles in the amplification step.

Quantification and quality control of the libraries were done using Qubit fluorimeter and Agilent 4200 TapeStation (Agilent, California, USA).

For library preparation (Cohort2 and Cohort5) we used the KAPA mRNA HyperPrep Kit (Kapa Biosystems, Wilmington, MA, USA).

### RNA sequencing

Single-read sequencing of the libraries with a read length of 75 was performed with NextSeq 500 Sequencing System using NextSeq 500/550 High Output v2 kit (75 cycles) (20024906, Illumina, San Diego, CA).

### RNA editing detection

Short-reads alignment of RNA-seq data was performed by BWA (ver. 0.6.2) [57] with default parameters, using the sacSer3 assembly (downloaded from the UCSC Genome Browser website [58]) as a reference genome. We augmented the genome with the coding sequence of the ADAR enzymes to allow alignment of ADAR-derived sequencing reads. Insert sizes for DNA- and RNA-seq samples were set to 700bp and 1,400bp respectively. A larger insert size was used for the RNA-seq samples to account for splicing events. Then RES-scanner [45] was applied for de-novo detection of RNA editing sites using default parameters, except for the following ones:—editLevel 0—editDepth 5—ss 0. The different RNA-seq replicates were separately compared to the BY4743 WT DNA-seq. As the RNA-seq procedure did not preserved the expressed strand information, mismatches of complement types (e.g. A-to-G and T-to-C, C-to-T and G-to-A etc.) could not be differentiated and thus mismatches were classified into 6 mismatch types (e.g. T-to-C mismatches and A-to-G mismatches are counted together). The sites identified by RES-scanner were further filtered by applying a Benjamini-Hochberg correction using the raw p-values reported by RES-scanner and setting the number of multiple tests conservatively to 12 million (genome size). The editing index was calculated as described in Roth et al [48], over the whole yeast genome.

The following site (http://degradome.uniovi.es/cgi-bin/nVenn/nvenn.cgi) was used to generate the Venn diagram shown in S1 Fig. For the logo motif we used: bedtools.2.27.1 and sacSer3 genome assembly.

### Secondary structures and their energies

To search for secondary structures that include the putative editing sites, we retrieved 801bp-long genomic sequence flanking each editing sites (400bp to each side) and used FOLD program from the RNAStructure package [59] with default parameters (except for temperature which was set to 30˚C = 303.15K) to find the lowest energy structure. Then, the substructure surrounding the editing site was extracted using bpRNA [60], and, if it was not contiguous, we connected its two arms by 7 "N" base-pairs. We recalculated the free energy (ΔG) of this

substructure using RNAStructure, taking the most probable structure. Sites which were not located within a double-stranded substructure were assigned $\Delta G = 0$.

## Supporting information

**S1 Fig. Distribution of detected sites (Fig 3A) across genomic regions.**
(PDF)

**S2 Fig. Overlap of editing sites identified in the different strains.**
(PDF)

**S3 Fig. Validating the editing activity of mdADAR1.**
(PDF)

**S4 Fig. Reproducibility of editing detection.**
(PDF)

**S5 Fig. Induced expression of mdADAR1 in human HeLa cells results in increased editing levels compared to the levels seen for overexpressed hADAR2A.**
(PDF)

**S1 Table. Yeast strains used in this study.**
(PDF)

**S2 Table. Plasmids used in this study.**
(PDF)

**S3 Table. RES-scanner results for mdADAR1, mdADAR1 E619A and URA empty.**
(CSV)

## Acknowledgments

The authors thank the Ben-Aroya and Levanon laboratory members for helpful comments on earlier versions of the manuscript.

## Author Contributions

**Conceptualization:** Adi Avram-Shperling, Amit Ben-David, Joshua J. C. Rosenthal, Erez Y. Levanon, Eli Eisenberg, Shay Ben-Aroya.

**Data curation:** Adi Avram-Shperling.

**Formal analysis:** Adi Avram-Shperling, Eli Kopel, Itamar Twersky, Orshay Gabay, Eli Eisenberg.

**Investigation:** Adi Avram-Shperling.

**Methodology:** Adi Avram-Shperling, Amit Ben-David, Sarit Karako-Lampert.

**Supervision:** Erez Y. Levanon, Eli Eisenberg, Shay Ben-Aroya.

**Writing – original draft:** Adi Avram-Shperling, Shay Ben-Aroya.

**Writing – review & editing:** Eli Eisenberg, Shay Ben-Aroya.

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
