## [Decision Letter · Decision Letter 0]

4 Sep 2022

Dear Dr Ben-Aroya,

Thank you very much for submitting your Research Article entitled 'Identification of exceptionally potent adenosine deaminases RNA editors from high body temperature organisms' to PLOS Genetics.

The manuscript was fully evaluated at the editorial level and by independent peer reviewers. The reviewers appreciated the attention to an important problem, but raised some substantial concerns about the current manuscript. Based on the reviews, we will not be able to accept this version of the manuscript, but we would be willing to review a much-revised version. We cannot, of course, promise publication at that time.

If you decide to revise the manuscript for further consideration at PLOS Genetics, please aim to resubmit within the next 60 days, unless it will take extra time to address the concerns of the reviewers, in which case we would appreciate an expected resubmission date by email to plosgenetics@plos.org.

[LINK]

We are sorry that we cannot be more positive about your manuscript at this stage. Please do not hesitate to contact us if you have any concerns or questions.

Yours sincerely,

Heather Hundley

Guest Editor

PLOS Genetics

Quanjiang Ji

Section Editor

PLOS Genetics

Overall, all three reviewers found the study of interest, that the manuscript was well-written and the experiments properly controlled. However, all three reviewers also noted that the data provided do not support the overall hypothesis. The two major points of concern in this regard are that the authors have not clearly demonstrated the dsRNA structures edited by the ADAR enzymes differ in stability and the differences in catalytic activity are due specifically to the changes in editing efficiency vs dsRNA binding or evolutionary fitness.

Reviewer's Responses to Questions

**Comments to the Authors:**

Reviewer #1: In this work, the authors used a yeast system to explore the efficiency of editing by ADAR enzymes taken from different organisms. They used yeast because it doesn’t contain ADAR enzymes and editing, making it a clean background for finding A-to-I changes. Testing several organisms, they found 3 to be very efficient in this setup. The most potent editor was from mallard-duck, which seems to be functional even in elevated temperatures. The two others that were tested seemed to be less efficient when the temperature rises, probably because that dsRNAs are less stable in elevated temperatures. They concluded that probably because mallard-duck natural temperatures are relatively high, ADARs in this organism evolved to bind weaker dsRNA structures.

The paper is well written and its approach and findings are interesting and worth publishing.

Although the findings are strong, the main weakness of the paper is that it is very speculative in concern of the dsRNA structures. Please see below the specific comments.

Specific comments:

1. It is unclear to me why in Figure 1D the authors focused on owADAR2 and not OWADAR1, which seems to be more imperative. Figure 1D should be better explained and why the authors chose to focus on these three proteins.

2. In which regions of the yeast genome editing occur? As yeast does not contain many introns and ORFs tend not to have dsRNA structures. It will be interesting to compare editing regions in the original organism to the new sites in yeast.

3. The hypothesis is that dsRNA structures are less stable at high temperatures and therefore are less edited. Specifically, in hADAR2 and hbADAR2. I would like to see specific examples. Concentrating on editing sites in more stable dsRNA structure, these sites should still be edited at high temperatures while less stable dsRNA should lose their editing. Showing the RNA folding experimentally would be great but this can be done bioinformatically by estimating the dG.

4. The last point is also important because of the difference in the preference of the editing enzymes, for example, hADR1 and hADAR2. It is possible that one enzyme (mallard) edits every site in a dsRNA (or even ssRNA) because of preference and not because of high temperatures and the others target specific structures that could change in variable temperatures.

5. I would like to see statistics (p-value) for Figure 5, specifically 5B. These are biological replicas and it shouldn’t be a problem.

Reviewer #2: In this manuscript, the authors applied the baker yeast as a system to evaluate the editing potential of ADARs from five species (two mammals and two birds, and one ectotherm invertebrate) that inhabit different environmental niches. They identified hummingbird and mallard-duck ADARs, which evolved at 40-42 degrees, as two exceptionally potent editors. They proposed that ADAR enzymes that target weaker dsRNA structures might be more effective than other ADARs and be used for Site-Directed RNA Editing.

Overall, this study proposes a novel system to identify better ADARs for RNA base editing. The major limitation of the current study is that the authors provide no direct evidence to support their hypothesis. Therefore, more experiments are needed to support their claims.

Major points

1. The authors propose that hummingbird and mallard-duck ADARs may target weaker dsRNA structures and therefore be more potent than other ADARs. Thus, the authors may compare the secondary structures surrounding the editing sites targeted by ADARs from the five species to confirm this claim.

2. For RNA base editing using exogenously expressed ADARs, we typically fused the ADAR’s catalytic domain with a λN peptide, a SNAP-tag, or a Cas protein (dCas13), and a guide RNA was designed to recruit the enzyme to the specific site. To achieve high on-target editing efficiency, we need a catalytic domain with high editing efficiency. In this study, it is unclear whether the high editing efficiency of hummingbird and mallard-duck ADARs is due to the less stringent structural requirement of their dsRNA binding domains or the highly active catalytic domains. Thus, the authors need to show that the catalytic domain of hummingbird or mallard-duck ADAR has high editing efficiency. e.g. a comparison of the editing efficiency between dCas13-human ADAR catalytic domain and dCas13- mallard-duck catalytic domain is essential for this study.

3. Since RNA base editing is mainly applied to mammalian systems, particularly in humans, the authors may express hummingbird and mallard-duck ADARs in human cells and examine their editing profiles/efficiencies.

Minor points

1. The editing sites may be better characterized. e.g. the editing level distribution and the genic location of the editing sites can be shown.

2. Figure 1A is somehow misleading. It seems that the authors inserted 5 genes into one plasmid.

Reviewer #3: As the authors note, use of ADAR RNA editing enzymes to alter mRNA sequences for therapeutic means is an active area of research, and optimizing editing efficiency at specific adenosines is a key goal in these endeavors. Since ADARs target dsRNA, which is less stable at higher temperatures, the authors propose that animals with higher core temperatures, or that live at higher temperatures, might have ADARs that target less stable dsRNA structures.

Prior studies by the Beal lab showed that increased ADAR editing in S. cerevisiae impairs cell growth, so in a clever use of this information, the authors express ADARs from endothermic vertebrates with different core body temperatures, or an ectothermic invertebrate that inhabits environments with different temperatures, in yeast, and use cell growth to screen for ADARs with different editing activities. Endogenous editing levels in the various yeast strains were determined with multiple assays that the authors have substantial expertise in from prior research. Indeed ADAR enzymes that cause the highest levels of editing (mallard duck (md) ADAR1, hummingbird (hb) ADAR2, orca whale (ow) ADAR2) are those that cause the most severe growth defects, thus validating the assay. For the most part these experiments are convincing and well-controlled, and obvious caveats like possible effects from altered protein levels are ruled out.

The authors next focus on mdADAR1 and hbADAR2, the enzymes that produce the greatest editing levels in yeast, and using human ADAR2 (hADAR2) as control, they perform the same experiments in yeast grown at different temperatures (25°C, 30°C, 34°C). As I understand it, the stated goal is to test the idea that mdADAR1 and hbADAR2 have higher editing levels because they can edit dsRNA that is less stable, and thus should have higher levels of editing than hADAR2 at higher temperatures. Although the authors see that temperature has a big effect on editing levels in yeast expressing hADAR2, there is little effect of temperature on editing when hbADAR2 or mdADAR1 are expressed, although the authors note in the Figure 5 legend:

“Res-scanner detection of editing sites (Methods) demonstrates a clear negative correlation between the growth temperature and editing activity for hADAR2 and hbADAR2.”

The data shown in Figure 5 do not support the above statement. Evaluating the data shown in Figure 5 with a student’s t-test is warranted, and may indeed support this statement, at least for hADAR2. That said, an effect of temperature on hADAR2 editing levels, without an effect on those from hbADAR2 or mdADAR1 might be expected if the authors’ hypothesis that the latter enzymes can edit dsRNA that is less stable is true. However, additional data would be necessary to support this hypothesis.

In summary, I am convinced that the authors have identified at least two ADARs that appear to be highly active, and thus possibly useful for improving therapeutic uses of ADARs, but the observations provide little support for the idea that the differential effects are because these ADARs are editing less stable structures. In this regard unless additional data are provided, the authors should tone-down statements about this, including the sentence in the Introduction that reads: “We provide evidence that these birds which evolved to live with higher core body temperatures have developed ADAR enzymes that target weaker dsRNA structures, and would therefore be more potent than other ADARs (compared at equal temperatures).”

Importantly, the authors already have data that could be mined to test their hypothesis. For example, the authors have mapped a huge number of editing sites, and if their hypothesis is correct, the editing sites unique to mdADAR1 shown in Supplementary Figure 1 would be predicted to occur in less stable structures. There are several algorithms that could make a start in determining whether these sites occur in less stable structures.

In truth, from the data shown, it is impossible to deconvolute changes in editing levels from changes that might occur because certain enzymes have evolved to have a specific optimal temperature. One way to specifically address this would be to choose one temperature and perform in vitro experiments with the various enzymes to compare their ability to edit several different dsRNAs designed to have different thermodynamic stabilities.

**Have all data underlying the figures and results presented in the manuscript been provided?**

Reviewer #1: Yes

Reviewer #2: Yes

Reviewer #3: Yes

PLOS authors have the option to publish the peer review history of their article (what does this mean?). If published, this will include your full peer review and any attached files.

Reviewer #1: No

Reviewer #2: No

Reviewer #3: No

---

## [Decision Letter · Decision Letter 1]

8 Feb 2023

Dear Dr Ben-Aroya,

We are pleased to inform you that your manuscript entitled "Identification of exceptionally potent adenosine deaminases RNA editors from high body temperature organisms" has been editorially accepted for publication in PLOS Genetics. Congratulations!

Yours sincerely,

Heather Hundley

Guest Editor

PLOS Genetics

Quanjiang Ji

Section Editor

PLOS Genetics

Comments from the reviewers (if applicable):

We thank the authors for addressing the reviewers' initial concerns and apologize for the delays in re-review. In particular, the addition of bioinformatic data regarding the thermodynamic stability of the secondary structures around the editing sites provided additional confidence in the overall conclusions of the manuscript.

Reviewer's Responses to Questions

**Comments to the Authors:**

Reviewer #1: I went through the revised manuscript, and I am satisfied that the authors answered all of the concerns raised by all reviewers. Notably, the addition of figure 3c, which shows the thermodynamic stability, strengthens the paper’s conclusion. I recommend acceptance of the manuscript.

Reviewer #2: I am satisfied with the revision and support the publication of this study.

**Have all data underlying the figures and results presented in the manuscript been provided?**

Reviewer #1: Yes

Reviewer #2: Yes

PLOS authors have the option to publish the peer review history of their article (what does this mean?). If published, this will include your full peer review and any attached files.

Reviewer #1: No

Reviewer #2: **Yes: **Rui Zhang

**Data Deposition**

http://datadryad.org/submit?journalID=pgenetics&manu=PGENETICS-D-22-00838R1

**Press Queries**

---

## [Editor Report · Acceptance letter]

1 Mar 2023

PGENETICS-D-22-00838R1 

Identification of exceptionally potent adenosine deaminases RNA editors from high body temperature organisms 

Dear Dr Ben-Aroya, 

We are pleased to inform you that your manuscript entitled "Identification of exceptionally potent adenosine deaminases RNA editors from high body temperature organisms" has been formally accepted for publication in PLOS Genetics! Your manuscript is now with our production department and you will be notified of the publication date in due course.

With kind regards,

Zsofia Freund

PLOS Genetics

On behalf of:
